# Aquaporin splice variation differentially modulates channel function during marine teleost egg hydration

Alba Ferré[1], François Chauvigné[2], Cinta Zapater[3], Roderick Nigel Finn[1,4], Joan Cerdà[1]*

1 Institute of Agrifood Research and Technology (IRTA)-Institute of Biotechnology and Biomedicine (IBB), Universitat Autònoma de Barcelona, Barcelona, Spain, 2 Institute of Marine Sciences, Spanish National Research Council (CSIC), Barcelona, Spain, 3 Institute of Aquaculture Torre de la Sal, Spanish National Research Council (CSIC), Castellón, Spain, 4 Department of Biological Sciences, University of Bergen, Bergen, Norway

* joan.cerda@irta.cat

**Data Availability Statement:** The full-length cDNAs for SaAqp1ab1-WT, HhAqp1ab1-WT and SsAqp1ab2-WT are available in GenBank under

## Abstract

Aquaporin-mediated oocyte hydration is a developmentally regulated adaptive mechanism that co-occurs with meiosis resumption in marine teleosts. It provides the early embryos with vital water until osmoregulatory systems develop, and in the majority of marine teleosts causes their eggs to float. Recent studies have shown that the subdomains of two water channels (Aqp1ab1 and Aqp1ab2) encoded in a teleost-specific aquaporin-1 cluster (TSA1C) co-evolved with duplicated Ywhaz-like (14-3-3ζ-like) binding proteins to differentially control their membrane trafficking for maximal egg hydration. Here, we report that in species that encode the full TSA1C, in-frame intronic splice variants of Aqp1ab1 result in truncated proteins that cause dominant-negative inhibition of the canonical channel trafficking to the plasma membrane. The inhibition likely occurs through hetero-oligomerization and retention in the endoplasmic reticulum (ER) and ultimate degradation. Conversely, in species that only encode the Aqp1ab2 channel we found an in-frame intronic splice variant that results in an intact protein with an extended extracellular loop E, and an out-of frame intronic splice variant with exon readthrough that results in a truncated protein. Both isoforms cause dominant-negative enhancement of the degradation pathway. However, the extended and truncated Aqp1ab2-type variants can also partially escape from the ER to reach the oocyte plasma membrane, where they dominantly-negatively inhibit water flux. The ovarian follicular expression ratios of the Aqp1ab2 isoforms in relation to the canonical channel are lowest during oocyte hydration, but subsequently highest when the canonical channel is recycled, thus leaving the eggs endowed with >90% water. These findings suggest that the expression of inhibitory isoforms of Aqp1ab1 and Aqp1ab2 may represent a new regulatory mechanism through which the cell-surface expression and the activity of the canonical channels can be physiologically modulated during oocyte hydration in marine teleosts.

accession numbers AY626938, MW960022 and DQ889223, respectively. The full-length transcripts of SaAqp1ab1_v1, SaAqp1ab1_v2, HhAqp1ab1_v1, SsAqp1ab2_v1 and SsAqp1ab2_v2 were deposited in GenBank under accession numbers OR498208, OR498209, OR498210, OR498211 and OR498212, respectively. All relevant data are within the paper and its Supporting information files.

**Funding:** This work was supported by the Spanish Ministry of Science and Innovation (MCIN/AEI/ 10.13039/501100011033), and the European Regional Development Fund (ERDF) "A way of making Europe" (European Union), Grant no. AGL2016-76802-R (to J.C.). A.F. was recipient of a predoctoral contract from Spanish MCIN (BES-2014-068745). R.N.F. was supported by the University of Bergen (Norway). The funders had no role in study design, data collection and analysis, decision to publish, or preparation of the manuscript.

**Competing interests:** The authors have declared that no competing interests exist.

## Introduction

Massive water uptake during meiosis resumption of marine teleost oocytes is an adaptive mechanism that produces highly hydrated eggs that float when released in the hyperosmotic seawater. The mechanism is thought to have evolved as an exaptation that evolved to provide the hyposmotic embryos with life-supporting water until osmoregulatory systems developed [1]. The hydration process involves the generation of an intracellular osmotic gradient via maturational yolk proteolysis and ion transport, and the coordinated temporal insertion of a water channel ortholog of mammalian AQP1 termed Aqp1ab in the oocyte plasma membrane [2–5]. The teleost-specific Aqp1ab (previously termed Aqp1o or Aqp1b) is a water-selective integral membrane channel that evolved via tandem duplication of the *aqp1aa* gene [3, 6–9].

Very recently, however, it has been shown that in fact two Aqp1ab-type channels (Aqp1ab1 and Aqp1ab2) evolved via tandem duplication within a teleost-specific aquaporin-1 cluster (TSA1C: *aqp1aa-aqp1ab2-aqp1ab1*) [10]. The existence of the TSA1C was not previously noted due to differential *aqp1ab*-type gene loss in many lineages. For example, anguillid eels only retain the *aqp1ab2* gene, while cyprinids, such as zebrafish, only retain the *aqp1ab1* gene and the majority of species that incubate their eggs internally such as seahorses and guppies have either lost or inactivated both genes [10]. Remarkably, however, functional forms of the Aqp1ab1 and -1ab2 channels are almost exclusively retained in marine species that spawn hydrated pelagic eggs (pelagophils), with a third of the highly diverse modern euacanthomorph teleosts retaining both copies [10]. RNA tissue profiling revealed that in contrast to other members of the piscine aquaporin superfamily [11, 12], only the Aqp1ab-type channels are highly accumulated in the ovaries of marine pelagophil species [7, 10]. A recent study further showed that the Aqp1ab-type channel subdomains co-evolved with Ywhaz-like (14-3-3ζ-like) binding proteins to differentially regulate their trafficking to avoid competitive occupancy of the same plasma membrane space in the oocyte and accelerate bulk water influx for maximal egg hydration [10]. In this regard, phosphorylation of C-terminal residues of Aqp1ab1 and -1ab2 promotes the binding of two teleost-specific proteins YwhazLa and YwhazLb, with Aqp1ab2 showing exclusive specificity for YwhazLb, to facilitate trafficking to the proximal region of the plasma membrane microvilli. A second phosphorylation in loop D in Aqp1ab1 traffics the channel to the distal region of the microvilli, thus avoiding competitive membrane space occupancy. A separate study then revealed that both neurohypophysial and paracrine vasopressinergic signaling systems integrate to regulate the Aqp1ab-type membrane trafficking in the oocyte via a common arginine vasopressin (Avp)-Avpr2aa receptor mediated induction of the protein kinase A (PKA) pathway [13].

During the course of these earlier studies we also observed that some non-canonical Aqp1ab-type variants were expressed in the ovary of certain teleost species. The expression of aquaporin splice variants has previously been observed for mammalian AQP0 [14] and AQP2 [15, 16] in some cell types under pathophysiological conditions, but also for AQP4 [17–19], AQP6 [20] and AQP8 [21] in normal tissues. Similarly, aquaporin splice variants have also been described in crustaceans [22, 23], insects [24], mollusks [25] and plants [26]. In some cases, plant or animal aquaporin splice variants can result in modified proteins with altered permeability properties [20, 26]. Conversely, various isoforms of mammalian AQP4 generated by alternative start codons or programmed translational readthrough can modulate supramolecular clustering in the plasma membrane and channel permeation [27, 28]. In other cases, truncated proteins are produced which can result in dominant-negative impedance of membrane trafficking of the canonical channels [14, 19]. Co-localization and functional studies with truncated AQP0-Δ213 and AQP4-Δ4 suggest that these isoforms gained dominant-negative repressive roles by trapping the canonical channels in the endoplasmic reticulum (ER)

through hetero-oligomerization and altered conformation, thereby preventing their trafficking to the plasma membrane [14, 19].

In this context, in the present study we sought to investigate whether splice variation of the Aqp1ab-type channels harnesses homologous structure-function modulations, and whether such modulations could regulate oocyte hydration in marine pelagophil teleosts.

## Results

### Reverse-transcriptase (RT)-PCR reveals new mRNA isoforms of teleost Aqp1ab channels

The *aqp1ab1* and *aqp1ab2* genes in the TSA1C are closely juxtaposed downstream of the *aqp1aa* gene and consist of four exons that in modern euacanthomorph species typically span 1.5–3 kb (Fig 1A–1C). To investigate whether teleosts express isoforms of these channels, we carried out a RT-PCR screening of selected adult tissues, including the ovary, in three representative marine pelagophil species, the spariform gilthead seabream (*Sparus aurata*), and the pleuronectiforms (flatfishes) Atlantic halibut (*Hippoglossus hippoglossus*) and Senegalese sole (*Solea senegalensis*), followed by DNA Sanger sequencing. For this analysis, we employed species- and paralog-specific oligonucleotide primers mapping to exon 2 and 3' UTR (*aqp1ab1*) or exon 2 and 4 (*aqp1ab2*) (Fig 1A–1C). In the seabream, the canonical *aqp1ab1* transcript and two other longer variants, designated as *aqp1ab1_v1* and *aqp1ab1_v2*, were amplified in the tissues tested, whereas only one mRNA corresponding to the canonical *aqp1ab2* was detected (Fig 1D). The seabream *aqp1ab1_v1* was detected in the ovary, testis and rectum, while the *aqp1ab1_v2* was present in the brain, gills, intestine, rectum and liver. In both cases, however, the expression levels of both transcript variants were much lower than those of the canonical transcript as judged by the relative intensities of the PCR products. Similarly, in Atlantic halibut, both canonical *aqp1ab1* and *aqp1ab2* transcripts could be amplified in all the tissues

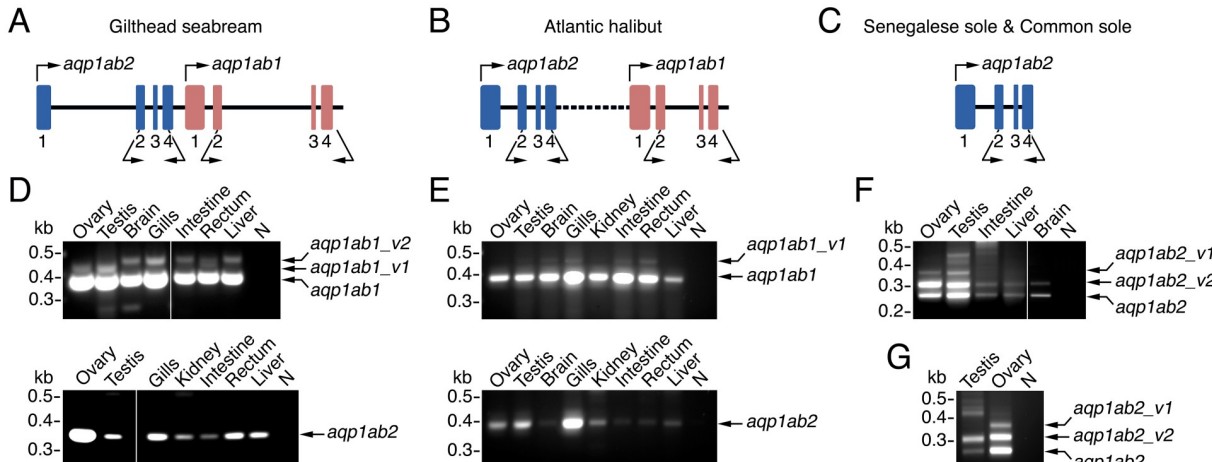

**Fig 1. Identification of *aqp1ab1* and *aqp1ab2* alternative splicing in teleosts.** (A-C) Genomic organization of tandemly arranged *aqp1ab2* and *aqp1ab1* genes in seabream (A) and Atlantic halibut (B), and of the single *aqp1ab2* gene in Senegalese sole and Common sole (C). The four exons (1–4) of both genes are represented by blue and red boxes, and the start codons are indicated by upper arrows. Lower arrows indicate the position of the oligonucleotide primers used for RT-PCR. (D-E) Representative RT-PCR analysis of *aqp1ab1* and *aqp1ab2* expression in different adult tissues from seabream (D) and halibut (E). A single *aqp1ab2* mRNA species is expressed in seabream and halibut, whereas two or one splice variants (*aqp1ab1_v1* and *aqp1ab1_v2*) in addition to *aqp1ab1* are detected, respectively, as indicated. (F-G) RT-PCR of *aqp1ab2* expression in Senegalese sole (F) and Common sole (G) where two splice forms (*aqp1ab2_v1* and *aqp1ab2 _v2*) in addition to *aqp1ab2* are amplified in the ovary and testis. In D-G, the sizes (kb) of molecular markers are indicated on the left.

analyzed, while one additional *aqp1ab1_v1* longer mRNA was detected at low levels in the brain, gills, intestine and rectum (Fig 1E). However, in the Senegalese sole, where *aqp1ab1* is absent in the TSA1C and a single *aqp1ab2* gene exists, two *aqp1ab2* mRNA variants longer that the canonical transcript were detected (Fig 1F). One of these isoforms (*aqp1ab2_v1*) was less expressed than the canonical transcript and was specific to the ovary and testis, while the other (*aqp1ab2_v2*) was equally or more abundantly expressed than the canonical transcript, and was amplified in all the tissues analyzed. Interestingly, both *aqp1ab2* variants were also detected in the ovary of the congener common sole (*S. solea*) (Fig 1G).

The full-length cloning and sequencing of the different aquaporin transcript variants revealed that seabream *aqp1b1_v1* and *aqp1ab1_v2* could potentially produce smaller Aqp1ab1 proteins (182 and 123 amino acids, respectively) than the canonical wild-type channel (SaAqp1ab1-WT, 267 amino acids), as a result of the in-frame addition of sequence fragments from intron 1 or 2, which introduce early stop codons and the production of truncated proteins missing exons 3 and 4 (SaAqp1ab1_v1) or exons 2, 3 and 4 (SaAqp1ab1_v2) (Fig 2A and 2D). Likewise, the Atlantic halibut *aqp1ab1_v1* mRNA includes an in-frame fragment of intron 3 after exon 3, resulting in the addition of a premature stop codon that would produce a shorter protein of 225 amino acids (HhAqp1ab1_v1), missing the complete exon 4, with respect to the HhAqp1ab1-WT (269 amino acids) (Fig 2B and 2E). In contrast, the Senegalese sole *aqp1ab2_v1* mRNA showed an in-frame addition of a proximal sequence of intron 2 after exon 2, which results in a larger protein with an extended extracellular loop E and four intact exons (SsAqp1ab2_v1) with respect to the SsAqp1ab2-WT (300 and 266 amino acids, respectively (Fig 2C and 2F). The sole *aqp1ab2_v2* variant showed the addition of a distal sequence of intron 2 also after exon 2, but in this case there is a frame change that continues through exon 3 and the beginning of exon 4 to end in a premature stop codon. This results in a smaller protein (SsAqp1ab2_v2) of 221 amino acids missing most of exon 4 (Fig 2C and 2F).

## Functional characterization of Aqp1ab1 and Aqp1ab2 isoforms and protein subcellular localization

The *in silico* three-dimensional structural analyses of the predicted Aqp1ab1 and Aqp1ab2 isoforms suggested that some of these variants had the general transmembrane helix structure intact (Fig 2D–2F). Therefore, we next investigated whether the different aquaporin isoforms could be functional using *Xenopus laevis* oocytes as an expression system. For this purpose, oocytes were injected separately with cRNAs coding for WT SaAqp1ab1, HhAqp1ab1 or SsAqp1ab2 channels, their splice isoforms, or with water as negative controls, and subsequently submitted them to swelling assays. Since Aqp1ab2 can only traffic to the frog oocyte plasma membrane when it is bound to the teleost-specific YwhazLb protein in the presence of the intracellular cAMP activator forskolin (FSK) [10], oocytes injected with SsAqp1ab2-WT or its isoforms were also co-injected with Atlantic halibut YwhazLb cRNA and exposed to FSK for 1 h prior to the determination of oocyte swelling. Oocytes expressing SaAqp1ab1-WT or HhAqp1ab1-WT showed a 5.2- and 5.6-fold increase in osmotic water permeability ($P_f$), respectively, with respect to the control oocytes after a hyposmotic challenge (Fig 3A and 3B), whereas oocytes injected with SsAqp1ab2-WT showed a $P_f$ increment of ~4 times (Fig 3C). However, with respect to the controls, oocytes expressing each of the seabream and halibut Aqp1ab1 or sole Aqp1ab2 isoforms did not show an increment of the oocyte $P_f$ (Fig 3A–3C).

To visualize the cellular distribution for each aquaporin isoform in *X. laevis* oocytes, and evaluate the possible mistargeting of these channels to the plasma membrane, oocytes were injected with untagged SaAqp1ab1-WT, FLAG-tagged HhAqp1ab1-WT or SsAqp1ab2-WT, or human influenza hemagglutinin (HA)-tagged splice variants. Previous experiments showed

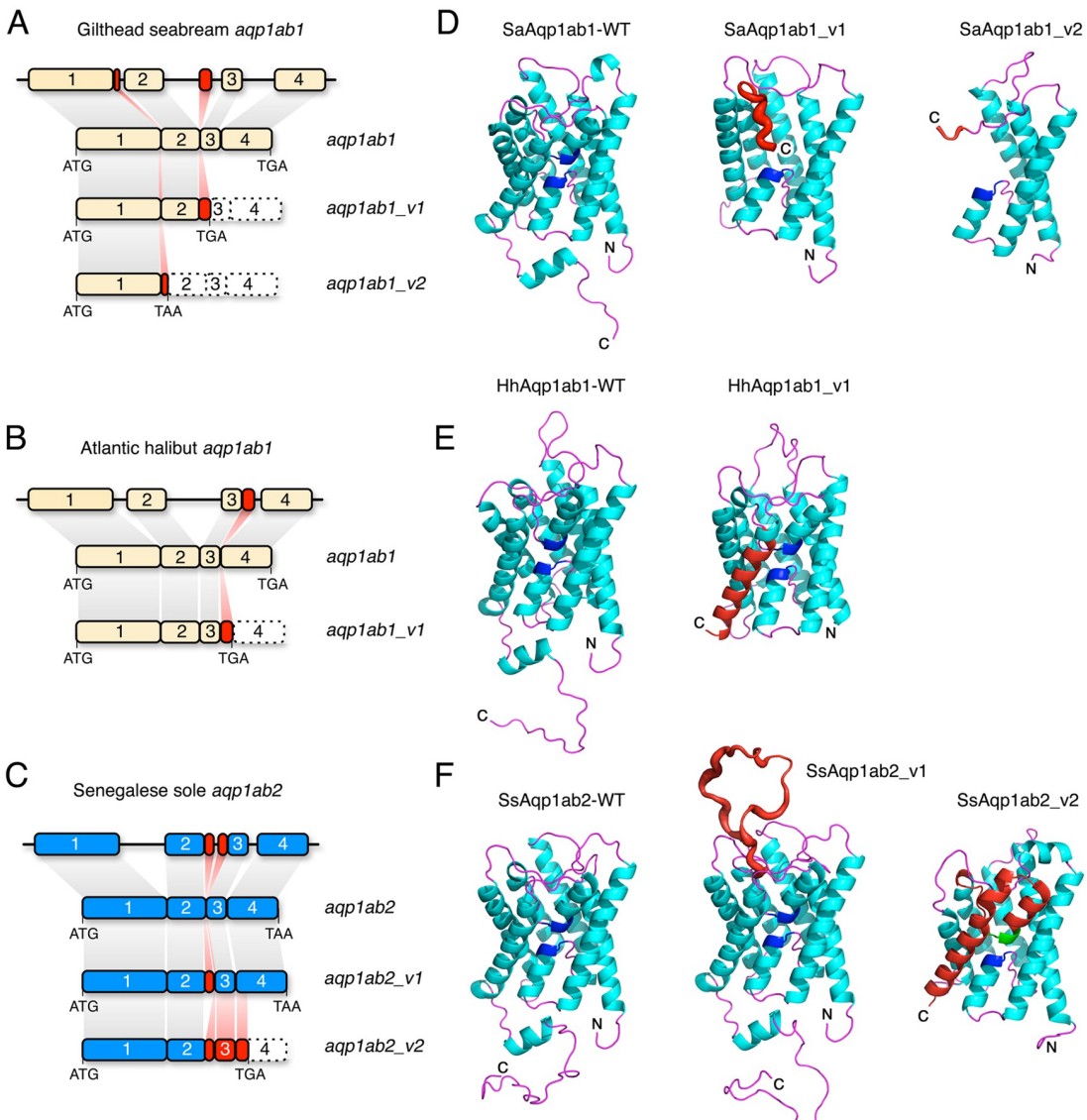

**Fig 2. Model-based structure of teleost Aqp1ab1 and Aqp1ab2 and splice variants.** (**A-C**) Schematic diagram of the seabream and halibut aqp1ab1 and sole aqp1ab2 genes, and the different emerging transcript variants identified in each species. The numbers represent the exons, and the initiation and stop codons in each mRNA are indicated. (**D-F**) Integral membrane views (cartoon renders) of the wild type and variant channels. Canonical transmembrane domains are shown in cyan, loops in magenta and the two conserved Asn-Pro-Ala (NPA) motifs in dark blue. Variant regions are shown in red. In sole Aqp1ab2_v2 (C and F), the splicing of the distal region of intron 2 causes out-of-frame readthrough of exon 3 and the spliced exon 4 resulting in a premature stop codon and the truncated protein.

that the Flag tag does not affect the function of the HhAqp1ab1 and SsAqp1ab2 channels (S1 Fig). Subsequently, we carried out double immunostaining using paralog-specific (SaAqp1ab1), or anti-FLAG (HhAqp1ab1 and SsAqp1ab2) or anti-HA (splice forms) antibodies, together with antibodies against the ER marker protein disulfide isomerase (PDI). These experiments showed that seabream and halibut Aqp1ab1-WT polypeptides were constitutively expressed at the oocyte plasma membrane, whereas their respective HA-SaAqp1ab1_v1, HA-SaAqp1ab1_v2 and HA-HhAqp1ab1_v1 isoforms only displayed intracellular staining

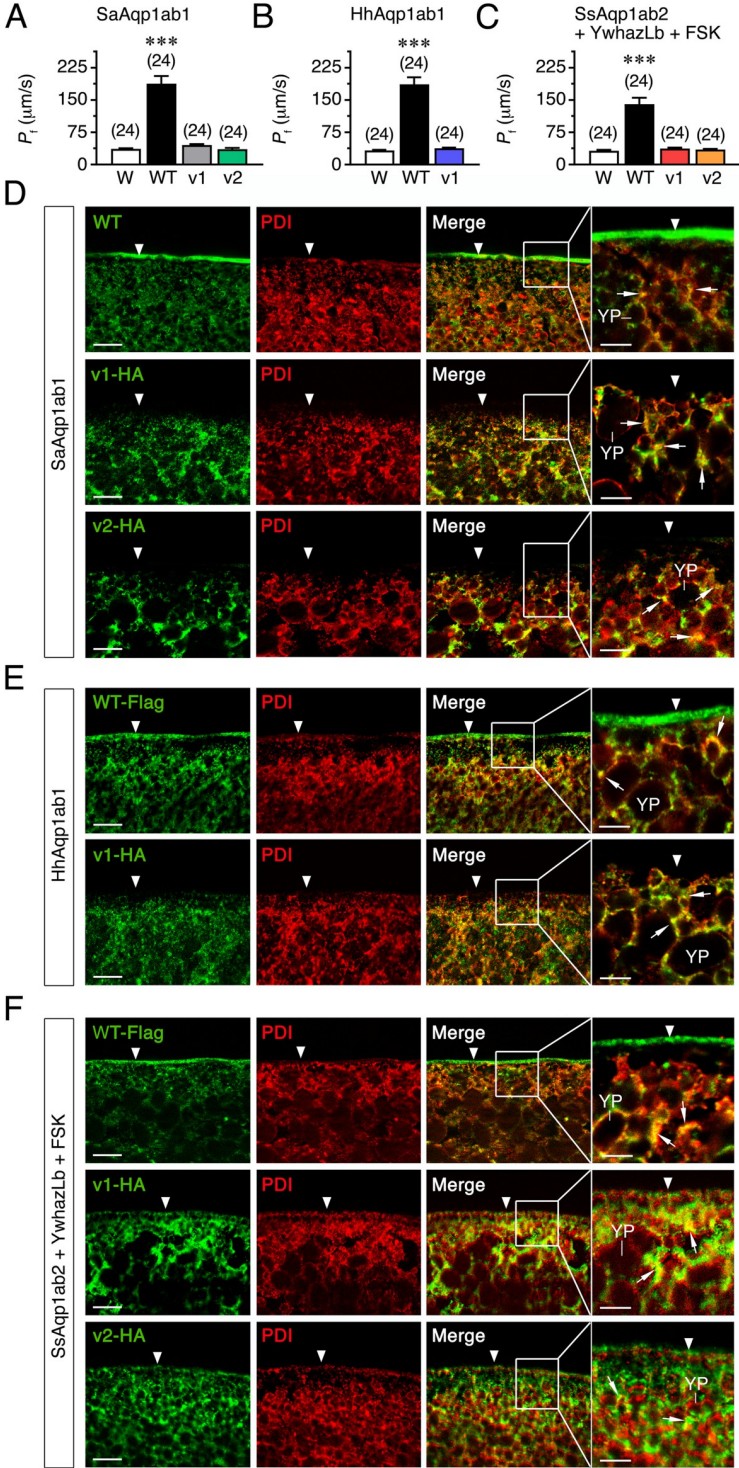

**Fig 3. Functional characterization and subcellular localization of teleost wild-type Aqp1ab1 and -1ab2 and their splice variants.** (**A-C**) $P_f$ of *X. laevis* oocytes injected with water (W, controls) or expressing seabream SaAqp1ab1-WT, SaAqp1ab1_v1 or SaAqp1ab1_v2 (A), Atlantic halibut HhAqp1ab1-WT or HhAqp1ab1_v1 (B), or sole SsAqp1ab2-WT, SsAqp1ab2_v1 or SsAqp1ab2_v2 (C). Oocytes injected with SsAqp1ab2-WT or splice forms were co-injected with halibut YwhazLb and exposed to FSK for 1 h prior to the swelling assay. The data are the mean ± SEM (*n* indicated above each bar) and were statistically analyzed by an unpaired Student's *t*-test (***, $P < 0.001$; with respect to controls). (**D-F**) Double immunostaining of oocytes expressing untagged SaAqp1ab1-WT or HA-tagged SaAqp1ab1_v1 or HA-SaAqp1ab1_v2 (D), Flag-tagged HhAqp1ab1-WT or HA-tagged HhAqp1_v1 (E), or Flag-tagged

SsAqp1ab2-WT or HA-tagged SsAqp1ab2_v1 or HA-SsAqp1ab2_v2 (F), and the ER marker protein disulfide isomerase (PDI). The plasma membrane is indicated by an arrowhead, whereas the co-localization of aquaporin channels and PDI in the cytoplasm is indicated by arrows. Scale bars, 10 μm (insets 5 μm).

(Fig 3D and 3E). These channel isoforms colocalized with the PDI, suggesting their impaired sorting from the ER to the frog oocyte plasma membrane (Fig 3D and 3E). In the presence of YwhazLb and FSK, the Flag-SsAqp1ab2-WT was also targeted to the plasma membrane, and most of the HA-SsAqp1ab2_v1 and HA-SsAqp1ab2_v2 signals colocalized with PDI in the cytoplasm (Fig 3F), as observed for the seabream and halibut aquaporin isoforms. However, HA-SsAqp1ab1_v1 and HA-SsAqp1ab2_v2 staining was also detected in the oocyte plasma membrane (Fig 3F), suggesting that the sole isoforms were not completely sequestered in the ER.

## Aqp1ab-type protein isoforms display different mechanisms of dominant-negative inhibition of the canonical channels

The previous immunolocalization data suggested that the identified aquaporin isoforms could have dominant-negative effects on cell surface expression of the full-length Aqp1ab proteins, and therefore impair channel function. To investigate this possibility, we expressed untagged SaAqp1ab1-WT, HhAqp1ab1-WT or SsAqp1ab2-WT, alone or in combination with different amounts of their respective non-tagged isoforms, in *X. laevis* oocytes, and subsequently determined the $P_f$. Water fluxes in oocytes expressing SaAqp1ab1-WT with increasing amounts of SaAqp1ab1_v1 or SaAqp1ab1_v2, in ratios from 0.5 to 50 (isoform:WT), were reduced in a dose-dependent manner up to 71 ± 3% and 57 ± 4%, respectively, with respect to oocytes expressing the SaAqp1ab1-WT alone (Fig 4A). Similar coexpression results were observed for the HhAqp1ab1_v1 isoform, although in this case the $P_f$ of oocytes coexpressing HhAqp1ab1_v1 and HhAqp1ab1-WT in ratios from 0.05 to 50 was inhibited up to 94 ± 4% with respect to oocytes injected with the canonical channel only (Fig 4B). Finally, water permeability of oocytes expressing SsAqp1ab2-WT plus YwhazLb and treated with FSK was also disrupted by coexpression with the SsAqp1ab2_v1 or SsAqp1ab2_v2 isoforms, the $P_f$ being highly inhibited (91 ± 3% and 84 ± 3%, respectively) already after injecting equal cRNA amounts of SsAqp1ab1-WT and channel variants (Fig 4C). These data indicated that the SsAqp1ab2 variants were ~40 times more inhibitory than the Aqp1ab1-type isoforms on the respective canonical channels. However, when SaAqp1ab1 and HhAqp1ab1 were coexpressed with YwhazLa and exposed to FSK, which increases the trafficking of these channels to the oocyte plasma membrane [10], the $P_f$ of oocytes was incremented, resulting in the enhancement of the percentage of inhibition produced by their respective splice forms (Fig 4D). Since in native seabream oocytes SaAqp1ab1 binds YwhazLa [10], these observations suggest that Aqp1ab1-type channels are more sensitive to low levels of splice variant coexpression under *in vivo* conditions, as observed for SsAqp1ab2.

To determine whether any of the Aqp1ab variants could physically interact with the respective canonical proteins and misroute their trafficking, we next performed double immunostaining of WT channels and their isoforms in oocytes expressing tagged constructs as above, followed by immunoblotting and co-immunoprecipitation experiments. Immunofluorescence analyses showed that SaAqp1ab1-WT and Flag-HhAqp1ab1-WT were both targeted to the plasma membrane, whereas in the presence of HA-SaAqp1ab1_v1 or HA-SaAqp1ab1_v2, or HA-HhAqp1ab1_v1, respectively, the expression of the WT channels was reduced in the oocyte surface, and the proteins colocalized with their respective isoforms within intracellular

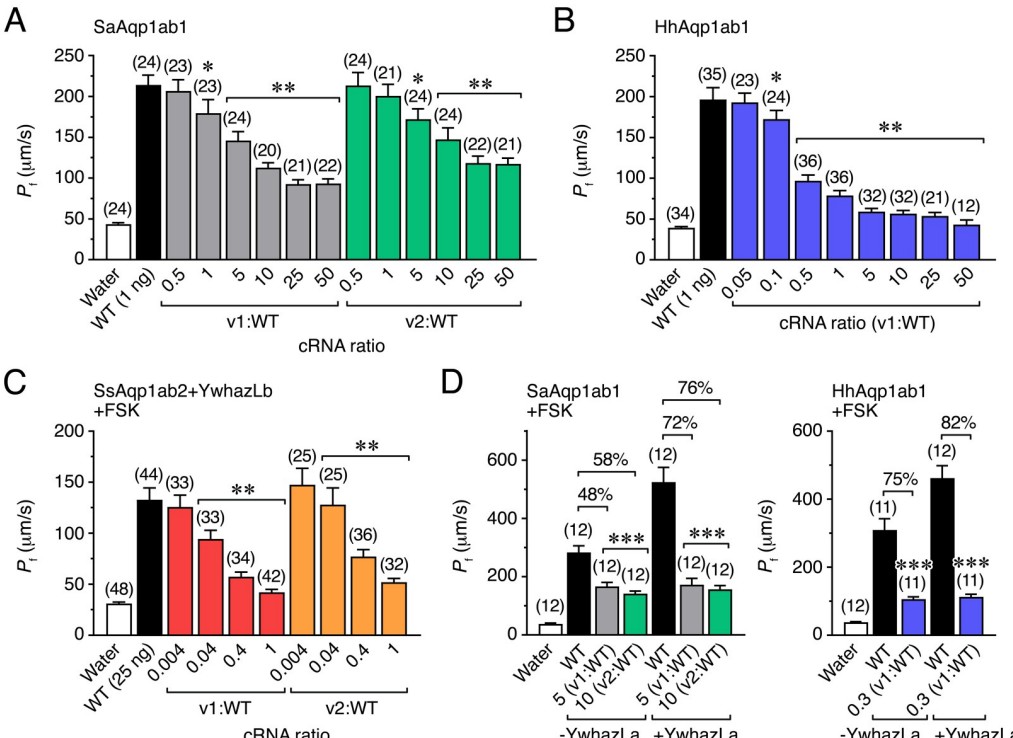

**Fig 4. The dominant-negative activity of teleost Aqp1ab1 and -1ab2 splice forms.** (**A-C**) Changes in $P_f$ of *X. laevis* oocytes injected with water or expressing SaAqp1ab1-WT (A), HhAqp1ab1-WT (B) or SsAqp1ab2-WT (C) alone or in combination with different amounts of the isoforms of each paralog. For SsAqp1ab2-WT and splice forms, oocytes were co-injected with halibut YwhazLb and exposed to FSK. (**D**) Effect of the inhibition of the oocyte $P_f$ by the different Aqp1ab1 splice forms, at a concentrations that produced approximately half-maximal reduction (as observed in A-C), in SaAqp1ab1 or HhAqp1ab1 expressing oocytes in the presence or absence of YwhazLa, and exposed to FSK. The percentage of $P_f$ inhibition elicited by each isoform under the two conditions is indicated in each plot. In all panels, data are the mean ± SEM (*n* indicated above each bar) and were statistically analyzed by an unpaired Student's *t*-test (*, $P < 0.05$; **, $P < 0.01$; ***, $P < 0.001$; with respect to oocytes injected with the WT form alone).

compartments (Fig 5A and 5B) compatible with ER as seen in previous experiments. Immuno-blotting of total and plasma membrane extracts of oocytes confirmed the absence of plasma membrane localization of the HA-SaAqp1ab1_v1 or HA-SaAqp1ab1_v2, and HA-HhAq-p1ab1_v1 isoforms, as well as the reduction of the relative amounts of SaAqp1ab1-WT and Flag-HhAqp1ab1-WT in the oocyte surface when coexpressed with their respective channel variants (Fig 5D and 5E). Co-immunoprecipitation trials using a SaAqp1ab1-specific antise-rum and a FLAG antibody (for HhAqp1ab1-WT) corroborated the interaction of the WT channels with their respective isoforms in the intracellular compartments (Fig 5G and 5H). These analyses also revealed that the interaction of SaAqp1ab1-WT with HA-SaAqp1ab1_v1, and that of Flag-HhAqp1ab1-WT with HA-HhAqp1ab1_v1, were associated with a decrease of the amount of the WT proteins in the oocyte total membrane fraction (Fig 5D and 5E) This suggested that multimeric complexes formed by the full-length and truncated isoforms are probably retained in the ER because they are incorrectly assembled and rapidly targeted for degradation. However, the complex formed by SaAqp1ab1-WT and HA-SaAqp1ab1_v2 appeared to be more stable, since the expression of the WT channel in the total membrane fraction of oocytes was not clearly affected by this truncated variant (Fig 5D).

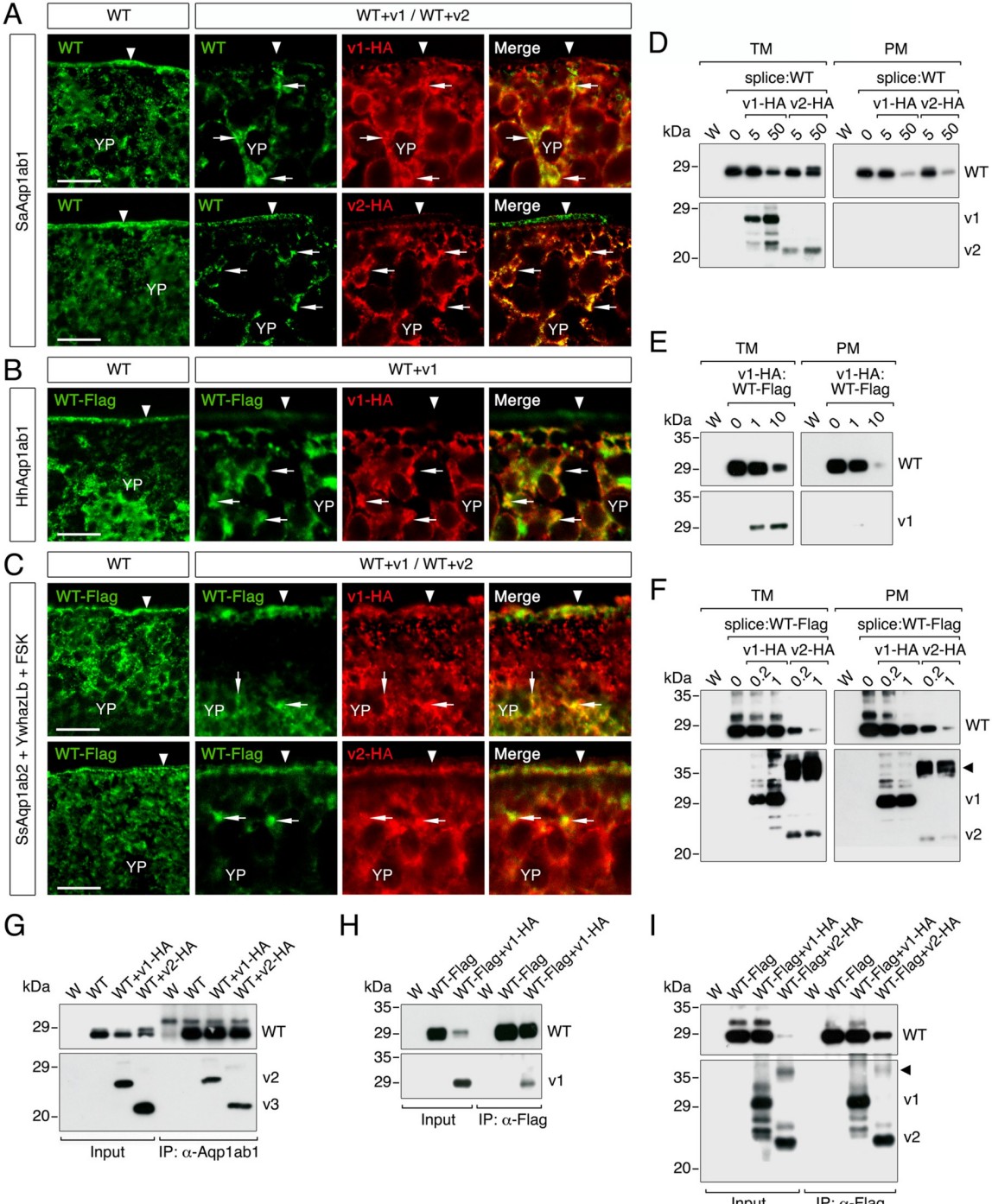

**Fig 5. Teleost Aqp1ab1 and -1ab2 splice forms generate distinct mechanisms of dominant-negative repression.** (**A-C**) Double immunolocalization of SaAqp1ab1-WT, Flag-tagged HhAqp1ab1-WT or Flag-tagged SsAqp1ab2-WT (green) with the respective HA-tagged splice forms in *X. laevis* oocytes as indicated. Oocytes expressing SsAqp1ab2-WT and splice forms were co-injected with halibut YwhazLb and exposed to FSK. In all panels, the plasma membrane is indicated by an arrowhead, whereas the co-localization of WT aquaporins and splice forms is indicated by arrows. Scale bars, 10 μm. (**D-F**) Representative immunoblots of total (TM) and plasma (PM) membrane protein extracts from oocytes injected with water (W) or co-expressing the different aquaporin WT channels with increasing amounts of the splice forms as indicated. (**G-I**) Illustrative immunoblots of precipitated proteins from oocytes treated as in A-C using anti-SaAqp1ab1 or anti-Flag (for HhAqp1ab1 and SsAqp1ab2) antibodies, and revealed with antibodies against SaAqp1ab1, Flag or HA. In D-I, upper blots were probed with SaAqp1ab1 (D and G) or Flag (E, F, H and I) antibodies, whereas in all panels the lower blots were probed with HA antibodies. Molecular mass markers (kDa) are on the left. The arrowhead in F and I indicate potential post-translational modifications of the SsAqp1ab1_v2 isoform.

In contrast to the seabream and halibut Aqp1ab1 channel variants, immunofluorescence and immunoblotting data indicated that the HA-SsAqp1ab2_v1 isoform with an extended extracellular E loop was targeted to the oocyte plasma membrane (Fig 5C), and did not affect the protein expression levels of the canonical channel (Fig 5F). However, the accumulation of the WT protein in the plasma membrane appeared to decline in the presence of high amounts of HA-SsAqp1ab2_v1 (Fig 5F), suggesting the progressive substitution of the WT channel by the inactive variant in multimeric tetramers (Fig 5I), which remain correctly assembled and are thus targeted to the oocyte surface. Interestingly, the HA-SsAqp1ab2_v2 isoform was highly post-translationally modified and also targeted to the oocyte plasma membrane, despite the truncated nature of this variant compared to the canonical channel (Fig 5C and 5F). In this case, however, interaction of HA-SsAqp1ab2_v2 with the Flag-SsAqp1ab2-WT induced the decrease of both the total and plasma membrane amounts of the WT channel (Fig 5F and 5I), suggesting that these complexes can partially escape from the ER and traffic to the oocyte surface.

## Expression pattern of Senegalese sole Aqp1ab2 isoforms during ovarian follicle growth and maturation *in vivo*

To investigate the physiological relevance of the Aqp1ab2 variants in Senegalese sole, we determined the *in vivo* dynamics of the endogenous channels during oocyte growth and maturation. For this purpose, two affinity-purified antibodies were prepared by immunizing rabbits with two synthetic peptides, one corresponding to a region from loop A conserved amongst the three SsAqp1ab2 proteins (α-Aqp1ab2-Nt), and another to a specific sequence of the SsAqp1ab2_v1 variant in loop E (α-Aqp1ab2_v1) (Fig 6A). The specificity of the antibodies was tested by Western blot analysis on total membrane protein extracts from *X. laevis* oocytes expressing SsAqp1ab2-WT, SsAqp1ab2_v1 or SsAqp1ab2_v2 separately. The results showed that the α-Aqp1ab2-Nt specifically recognized the three SsAqp1ab2 polypeptides, whereas the α-Aqp1ab2_v1 only reacted with the SsAqp1ab2_v1 protein product (Fig 6B). The antibodies revealed immunoreactive bands of ~28 kDa, ~32 kDa and ~24 kDa, thus approximately of the same molecular masses of the SsAqp1ab2-WT, SsAqp1ab2_v1 or SsAqp1ab2_v2 monomers predicted *in silico* (28.5, 32.2 and 23.6 kDa, respectively).

The changes in protein expression of the three SsAqp1ab2 variants were estimated by immunoblotting in isolated ovarian follicles at different developmental stages using both of the generated antisera. The stages analyzed were: previtellogenic; follicles enclosing oocytes at the primary growth and cortical alveoli stages; vitellogenic follicles, with oocytes filled with yolk granules containing vitellogenin-derived yolk proteins; hydrating follicles, with oocytes undergoing meiotic maturation and hydration; mature, follicle-enclosed oocytes that resumed meiosis and hydration; and ovulated eggs, free of the surrounding follicle cells (Fig 6C). The α-Aqp1ab2-Nt detected the protein products for all three SsAqp1ab2 variants in protein extracts from follicles in most developmental stages, which were identified by their apparent molecular mass (Fig 6D, upper left panel). However, several higher molecular bands were also revealed at the previtellogenic, vitellogenic and hydrating stages, which could represent post-translational modifications of the channels, since these bands, as well as those corresponding to the monomers, were no longer detected after the preadsorbtion of the antibody with the immunizing peptide (Fig 6D, upper right panel). The relative amounts of the SsAqp1ab2-WT slightly decreased from the previtellogenic to the hydrating stage, whereas those of the SsAqp1ab2_v1 and SsAqp1ab2_v2 variants showed a more drastic reduction during the same developmental stages (Fig 6D, upper left panel). In the mature and ovulated samples, SsAqp1ab2-WT immunoreactivity was no longer detected, while that of SsAqp1ab2_v1 also disappeared but only in

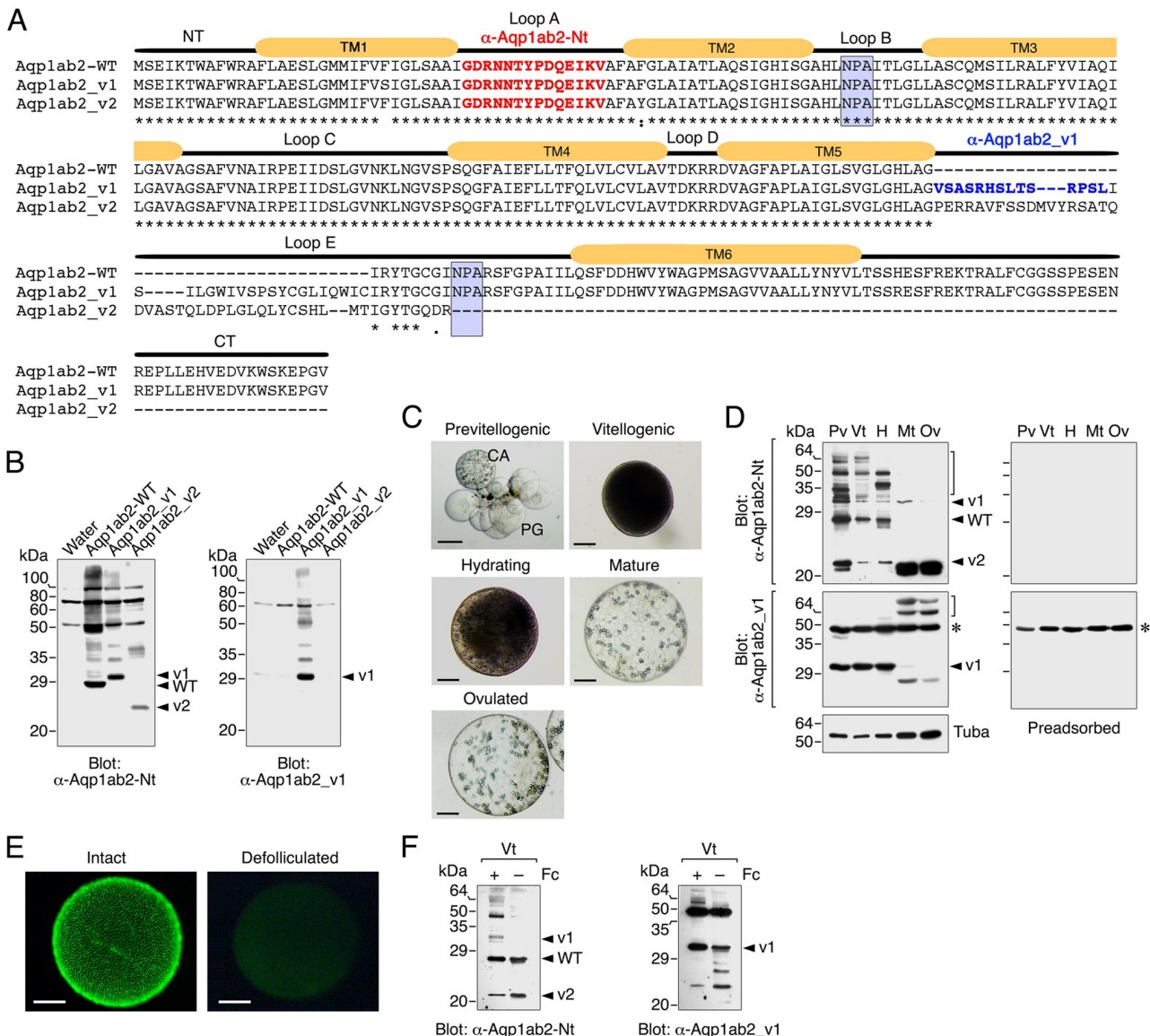

**Fig 6. Production of Senegalese sole total Aqp1ab2 and Aqp1ab2_v1 specific antisera and immunoblot analysis of Aqp1ab2 splice variants protein expression in developing ovarian follicles.** (**A**) Amino acid sequence alignment of SsAqp1ab2-WT, SsAqp1ab2_v1 and SsAqp1ab2_v2 splice forms highlighting the sequences employed for the production of the sole-specific Aqp1ab2 and Aqp1ab2_v1 antibodies (α-Aqp1ab2-Nt and α-Aqp1ab2_v1, in red and blue color, respectively). The N- and C-termini of the WT channel, the six transmembrane helices (TM1-TM6) and the two conserved NPA motifs are also indicated. (**B**) Immunoblots of *X. laevis* oocytes injected with water or expressing Aqp1ab2-WT, SsAqp1ab2_v1 or SsAqp1ab2_v2 and probed with the α-Aqp1ab2-Nt or α-Aqp1ab2_v1 antisera separately. (**C**) Representative photomicrographs of sole ovarian follicles at different developmental stages during oocyte growth and maturation. PG, primary growth stage; CA, cortical alveoli stage. Scale bars, 50 and 250 μm. (**D**) Immunoblots of Aqp1ab2-WT, Aqp1ab2_v1 and Aqp1ab2_v2 in protein extracts from previtellogenic (Pv), vitellogenic (Vt), hydrating (H) and mature (Mt) follicle-enclosed oocytes, as well as from ovulated oocytes (Ov), using the generated antisera. Alpha tubulin (Tuba) was used as loading control. Duplicated blots (right) were run in parallel and incubated with the primary antibodies preadsorbed by the antigenic peptide to test for specificity. The brackets indicate potential post-translational modifications. (**E**) Hoechst staining of intact vitellogenic follicles and defolliculated oocytes. Scale bar, 100 μm. (**F**) Immunoblots of Aqp1ab2 isoforms in total protein extracts from intact or defolliculated vitellogenic follicles (+/- follicle cells, Fc). In B, D and E, the arrowheads indicate aquaporin monomers, whereas the asterisks in D and E indicate cross-reactive polypeptides revealed with the Aqp1ab2_v1 antiserum. In all blots, molecular mass markers (kDa) are on the left.

ovulated eggs. In contrast, the SsAqp1ab2_v2 signal intensity strongly increased in both mature and ovulated eggs (Fig 6D, upper left panel).

The α-Aqp1ab2_v1-specific antibody revealed two major immunoreactive bands, one of ~32 kDa, approximately matching the molecular mass of the SsAqp1ab2_v1 isoform, and another of ~49 kDa, the intensity of which did not change throughout the different follicular stages (Fig 6D, lower left panel). The cross-adsorption of the antibody with its antigenic peptide eliminated the SsAqp1ab2_v2 corresponding band, but not the 49-kDa reactive band, indicating that the latter corresponds to a cross-reactive protein present in the sole ovarian follicles (Fig 6D, lower right panel). The SsAqp1ab2_v1 immunoreactivity in the follicular extracts did not change from the previtellogenic to the hydrating stage, but it was no longer detected in mature and ovulated eggs (Fig 6D, lower left panel). Instead, at these later stages, secondary reactive bands of higher and lower molecular masses than the predicted monomer were revealed (Fig 6D, lower left panel). These new bands could correspond to complex post-translational modifications and degradation products of SsAqp1ab2, respectively.

In our recent studies, we found that the seabream Aqp1ab2 paralog is not only expressed in the oocyte but also in the surrounding follicle cells [10, 13]. To assess whether the sole Aqp1ab2 and channel variants are expressed in the oocyte, we carried out immunoblotting of protein extracts from isolated intact vitellogenic ovarian follicles and defolliculated oocytes using the α-Aqp1ab2-Nt and α-Aqp1ab2_v1 antisera (Fig 6E). In agreement with the previous data, SsAqp1ab2-WT, SsAqp1ab2_v1 and SsAqp1ab2_v2 protein products were detected in extracts from intact follicles with the α-Aqp1ab2-Nt antibody, while in defolliculated oocytes the SsAqp1ab2_v1 isoform was not revealed (Fig 6F, left panels). In contrast, using the α-Aqp1ab2_v1-specific antiserum, the SsAqp1ab2_v1 variant became visible in both intact follicles and defolliculated oocytes (Fig 6F, right panels), suggesting that this antiserum is able to detect lower amounts of the SsAqp1ab2_v1 isoform than the α-Aqp1ab2-Nt antibody. These data therefore corroborate that the sole Aqp1ab2 paralog and its splice variants are expressed in vitellogenic oocytes as observed for the seabream Aqp1ab2 channel.

To confirm the previous observations and investigate potential changes in the subcellular localization of SsAqp1ab2 and the SsAqp1ab2_v1 isoform in ovarian follicles during oocyte growth and maturation, we conducted immunofluorescence microscopy on sole ovarian sections containing follicles at different stages. In previtellogenic follicles containing primary growth oocytes, total SsAqp1ab2 and SsAqp1ab2_v1-specific staining was dispersed in the oocyte cytoplasm, but positive signals were also found in the developing follicle cells (Fig 7A–7C). When oocytes enter into secondary growth and reach the cortical alveoli stage, total SsAqp1ab2 and SsAqp1ab2_v1 immunoreactions become accumulated towards the cortical region of the oocyte just below the plasma membrane (Fig 7D–7F). At this stage, differentiated granulosa and theca cells also showed positive staining with both antibodies (Fig 7D–7F). However, in fully-grown vitellogenic follicles, while follicle cells remained labelled by either of the antisera (Fig 7G–7I), total SsAqp1ab2 immunostaining was observed in the oocyte plasma membrane and partially distributed along the microvilli crossing the vitelline envelope (Fig 7H). In contrast, SsAqp1ab2_v1 signals remained accumulated in the oocyte cortex and plasma membrane (Fig 7I). In hydrating follicles, follicle cells were either not or slightly stained with the α-Aqp1ab2 and α-Aqp1ab2_v1 antisera, respectively (Fig 7J–7L). In addition, total SsAqp1ab2 staining almost disappeared from the oocyte cytoplasm and became accumulated in the microvilli (Fig 7K). On the contrary, that of SsAqp1ab2_v1 was detected in the cortical ooplasm but not in the oocyte plasma membrane (Fig 7L). In all stages, the specificities of the immunoreactions were demonstrated by the lack of staining when the antisera were preincubated with their corresponding antigenic peptide (S2 Fig). These data therefore indicated that the expression of SsAqp1ab2 and SsAqp1ab2_v1 is not restricted to the oocyte but also occurs in follicle

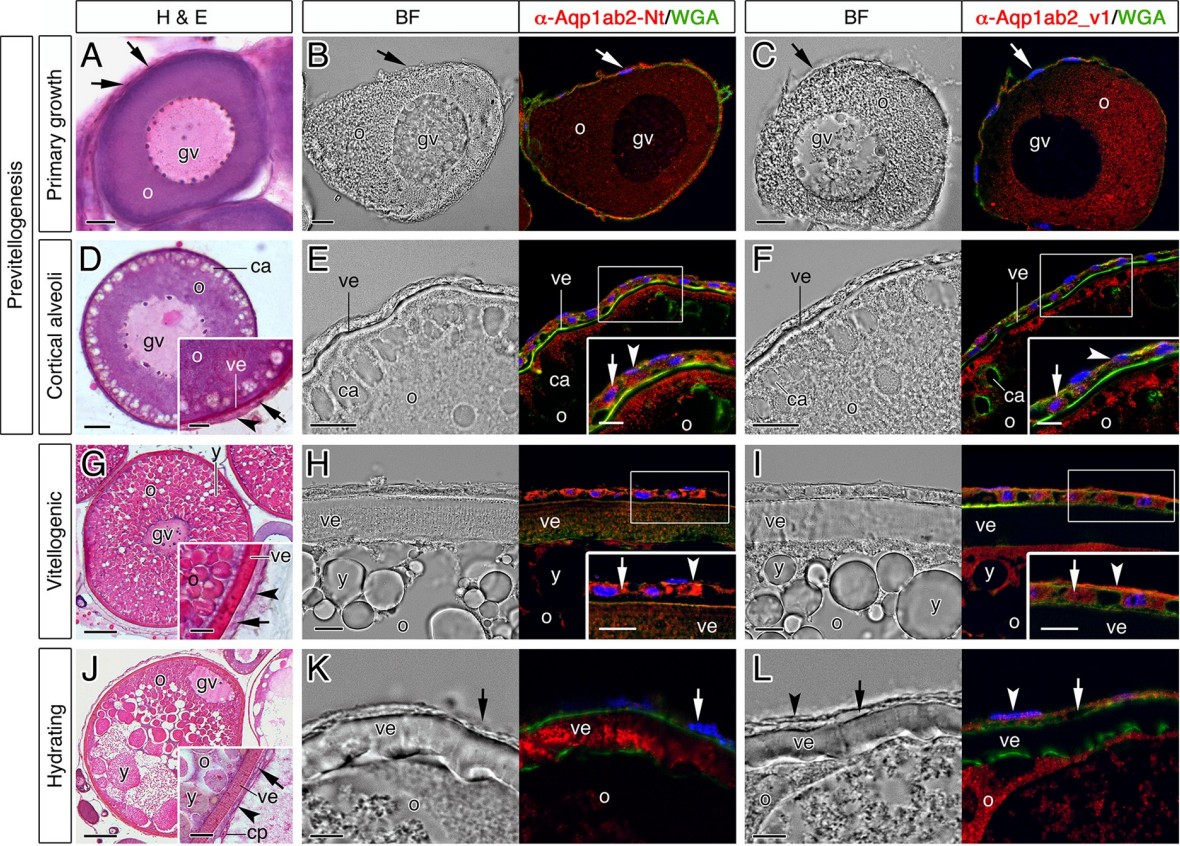

**Fig 7. Immunolocalization of total Aqp1ab2 and the Aqp1ab2_v1 isoform in developing Senegalese sole ovarian follicles. (A, D, G. J)** Representative histological sections of follicle-enclosed oocytes at previtellogenesis, including the primary growth (A) and cortical alveoli (D) stages, vitellogenic (G) and maturing and hydrating (J) stage stained with hematoxylin and eosin. (**B-C, E-F, H-I, K-L**) Bright field (BF) images and immunostaining of total Aqp1ab2 or Aqp1ab2_v1 (red) in follicles in each of the different stages using the sole-specific α-Aqp1ab2-Nt and α-Aqp1ab2_v1 antisera. Sections were counterstained with 4′,6-diamidino-2-phenylindole (DAPI; blue) and wheat germ agglutinin (WGA; green). Control sections incubated with preabsorbed antisera were negative (S1 Fig). The arrows and arrowheads, indicate granulosa and theca cells, respectively, surrounding the oocyte. Abbreviations: o, oocyte; y, yolk globule; gv, germinal vesicle; ve, vitelline envelope; cp, capillary; ca, cortical alveoli. Scale bars, 10 μm (A-D, H, I, L, K, and inset in I and L), 50 μm (G, J), 20 μm (E, F), 5 μm (inset in E and F).

cells, and that the trafficking of these channels to the oocyte plasma membrane is differentially regulated during hydration.

### Follicular Aqp1ab2 isoforms are differentially regulated during sole oocyte maturation and hydration *in vivo*

To analyze whether the expression levels of the SsAqp1ab2_v1 and SsAqp1ab2_v2 isoforms in relation to those of the SsAqp1ab2-WT could be developmentally regulated, we evaluated protein abundance of the three channels in three different pools of ovarian follicles at previtellogenesis, vitellogenesis and hydration, and in mature follicles, by immunoblotting using the α-Aqp1ab2-Nt antisera. The results showed that the relative amount of SsAqp1ab2_v1 with respect to that of the WT channel progressively decreased from previtellogenesis to hydration, whereas this ratio increased by ~10 times at the maturation stage (Fig 8A and 8B). The SsAqp1ab2_v2 to WT expression ratio also showed a decrease during vitellogenesis, but a small but significant increase was seen during hydration (Fig 8A and 8C). However, a stronger

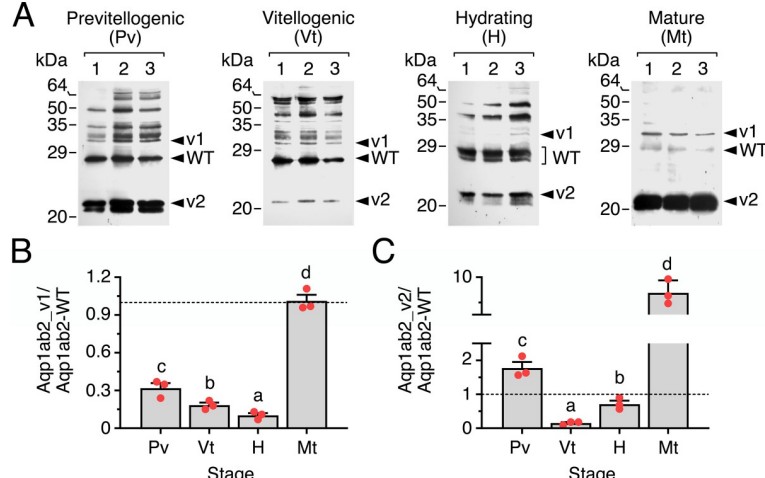

**Fig 8. Differential regulation of Aqp1ab2 splice variants during ovarian follicle maturation and hydration *in vivo*.**
(**A**) Representative immunoblots of protein extracts from follicles at the previtellogenic, vitellogenic), hydrating, and mature stage (2.5 follicle equivalents/lane) using the α-Aqp1ab2-Nt antisera. The blots show extracts from three different pools of follicles collected from three different females. The arrowheads indicate the Aqp1ab2-WT, Aqp1ab2_v1 and Aqp1ab2_v2 proteins according to their apparent molecular masses. In all blots, molecular mass markers (kDa) are on the left. (**B**) Relative amount of the Aqp1ab2_v1 (left) and Aqp1ab2_v2 (right) isoforms with respect to the Aqp1ab2-WT at the different development stages based on the blots shown in A. Bars (mean ± SEM; $n = 3$ fish) with different superscript are statistically significant (one-way ANOVA; $P < 0.05$).

increment (~10 times) of the amount of SsAqp1ab2_v2 with respect to that of the WT occurred at the maturation stage (Fig 8A and 8C). Thus, low protein expression levels of SsAqp1ab2_v1 and SsAqp1ab2_v2 during the hydration stage could facilitate SsAqp1ab2-WT trafficking to the oocyte plasma membrane and water influx, whereas the diminished SsAqp1ab2-WT protein levels in the mature follicles are likely to be the result of the negative effect of increased expression of SsAqp1ab2_v2.

## Discussion

In the present study, we identified five novel splice variants of *aqp1ab*-type genes from four species of marine pelagophil teleosts. For the Atlantic halibut and seabream, the splicing of intronic regions of the *aqp1ab1* genes introduces premature stop codons and thus C-terminally truncated proteins are translated. By contrast, in the Senegalese sole, the alternative in-frame splicing of the proximal second intronic region of the *aqp1ab2* gene generates an intact Aqp1ab2_v1 protein with an extended extracellular loop E, while the splicing of the distal region of the second intronic region leads to out-of-frame readthrough of exon 3 and 4 and the generation of a C-terminally truncated Aqp1ab2_v2 protein due to a premature stop codon. These *aqp1ab2_v1* and *-1ab2_v2* splice variants were also identified in the ovary of the common sole indicating that the isoforms are at least conserved within the genera. However, in contrast to the WT Aqp1ab1 and Aqp1ab2 channels, none of the variants were found to function as water channels when expressed in *X. laevis* oocytes, indicating that their positive selection in the different species evolved for a different purpose.

An insight into this potential purpose arose from the observations that the WT channels are constitutively localized in the plasma membrane when heterologously expressed in *X. laevis* oocytes, while the seabream and halibut Aqp1ab1 splice variants are retained in the ER. Conversely, the sole Aqp1ab2 splice isoforms are targeted to the oocyte plasma membrane

(Aqp1ab2_v1) or partially sequestered in the ER (Aqp1ab2_v2). By co-expressing each of the variants with the WTs and immunologically locating their proteins in the frog oocytes, it became evident that the splice variants elicit distinct mechanisms of dominant-negative inhibition of the WT channel function. Our double immunostaining and co-immunoprecipitation experiments suggest that one of these mechanisms, displayed by the three Aqp1ab1 variants and partially by the sole Aqp1ab2_v2 isoform, is gained by trapping the WT channels in the ER by hetero-oligomerization, thereby preventing their trafficking to the plasma membrane. This dominant-negative effect could be explained by the oligomeric structure of Aqp1ab1 and Aqp1ab2 tetramers, which are probably retained in the ER when they are formed by WT channels and truncated isoforms, because they are incorrectly assembled and thus targeted for degradation [19, 29]. Such a mechanism is therefore reminiscent of the dominant-negative effect caused by the truncated human AQP4-Δ4 and AQP0-Δ213 splice forms, which heterodimerize with the canonical channels to cause their retention in the ER [14, 19]. However, our data also revealed that the sole Aqp1ab2 splice variants, which show high inhibitory efficiency, are also targeted to the plasma membrane together with the Aqp1ab2-WT channel. Due to the lack of functionality of these variants as water transporters, possibly because of altered structural domains that affect the water pore or impede the formation of functional homotetramers in the plasma membrane [30], their increased trafficking to the plasma membrane results in dominant-negative repression of water permeation. These findings suggest that Aqp1ab-type splice variation in teleosts may have evolved to differentially and more efficiently regulate channel function in species that lost one of the *aqp1ab*-type genes, such as soleid fishes, which lost the *aqp1ab1* gene [10].

Currently, nothing is known concerning the physiological relevance of aquaporin splice variation in fishes. In the ovaries of seabream and halibut, the *aqp1ab1*-type variants are much lower expressed *in vivo* than the WTs, and therefore their biological significance is uncertain, although increased transcription and translation of these isoforms might occur due to stress associated with disease or in response to changes in the environment [31–34]. On the contrary, since the *aqp1ab2* splice variants are relatively highly expressed in the ovaries of soles, we aimed to determine their potential physiological significance *in vivo* during oocyte growth and maturation in the Senegalese sole. The immunological data show that sole Aqp1ab2-WT, Aqp1ab2_v1 and Aqp1ab2_v2 proteins are consistently expressed between previtellogenesis and oocyte hydration, but in fully mature and ovulated eggs the WT is strongly downregulated, while Aqp1ab2_v1 and Aqp1ab2_v2 expression continues, with levels of Aqp1ab2_v2 greatly increased. In terms of ratios to the WT, both of the splice variants show a general decrease as between previtellogenesis and hydration, but a substantial increase in the mature eggs. The immunostaining data also show that as for seabream Aqp1ab2-WT [10, 13], the sole Aqp1ab2-WT and its splice variants are expressed both in the theca and granulosa cells as well as in the oocyte. However, in contrast to the Aqp1ab2-WT, which begins to enter the oocyte plasma membrane microvilli during vitellogenesis, and is almost completely translocated there during hydration, the Aqp1ab2_v1 never enters the microvillar membrane. It would thus seem apparent that the roles of the splice variants in the follicular cells would be to modulate the rate of water flow during oocyte growth via the observed dominant-negative repression mechanisms. During hydration, their presence in the oocyte is strongly reduced so as not to inhibit maximal water influx through the canonical channel, which can thus form homotetramers and be fully translocated to the membrane microvilli. For species that retain both Aqp1ab1 and Aqp1ab2 in the TSA1C, such a regulation may not be under the same selection pressure as the two channels evolved to occupy the distal and proximal regions of the oocyte microvilli, respectively, thereby avoid competitive membrane space occupancy to accelerate bulk water influx during oocyte maturation [10, 13]. In the seabream, the differential trafficking of the

Aqp1ab-type paralogs in the oocyte is regulated via the luteinizing (Lh) hormone through a common Avp signaling pathway [13]. In sole, although the downregulation of the Aqp1ab2 splice variants occurs mainly during oocyte maturation and hydration, a process typically controlled by Lh in fishes [35], the endocrine pathway regulating the expression of these isoforms at this stage is unknown and should be investigated in the future.

In order for the hydration process to terminate and keep the acquired water within the mature egg, it has been shown that euacanthomorph teleosts evolved a C-terminal p38 MAPK site within the YwhazLa binding site of Aqp1ab1 that causes the channel to be recycled [7, 10]. However, since soleid flatfishes lack the *aqp1ab1* gene, the present data indicate that Aqp1ab2 recycling in mature oocytes is possibly promoted by increasing the levels of the Aqp1ab2_v1 and Aqp1ab2_v2 variants, which dominantly negatively enhance the degradation pathway.

Taken together, the present findings provide new evidence that euacanthomorph species of marine teleosts evolved an array of mechanisms to control aquaporin-mediated oocyte hydration. This includes the selective retention of the Aqp1ab-type channels which co-evolved subdomain phosphorylation sites with YwhazL binding proteins to differentially localize the channels in the oocyte membrane microvilli for maximal hydration [10, 13]. Here, we find that Aqp1ab-type splice variation adds a further layer of control through distinct mechanisms of dominant-negative regulation of channel function. This includes the inhibition of transmembrane water conductance due to the trafficking of non-functional variants to the plasma membrane, and the repression of the trafficking of the canonical channels, so that degradation pathways are enhanced. In marine pelagophil species that retain only one of the *aqp1ab*-type genes, the regulatory repression mechanisms achieved through splice variation may be relatively more important for the maintenance of hydration than in species that encode the complete TSA1C.

## Materials and methods

### Animals and sampling

Adult seabream and Atlantic halibut were obtained from aquaculture stations in Spain and Norway, respectively, and maintained as previously described [4, 36]. Two-year old adult Senegalese sole females were obtained from the commercial company Stolt Sea Farm S.A. (Spain), and transported and raised at the facilities of the Institute of Marine Sciences of Barcelona (Spanish Council for Scientific Research, CSIC, Spain). Fish were kept in 2000 l tanks under natural conditions of photoperiod and temperature, and fed with dry pellets (balanced diet, LE-7-Elite, Skretting) five days a week [37]. Tissue biopsies were collected from sacrificed fish during the natural spawning season for each species as previously described [4, 36]. For this, fish were sedated with 60 mg/l tricaine methanesulfonate (MS-222; Merck) and immediately sacrificed by decapitation. Tissue samples were frozen in liquid nitrogen and stored at -80˚C. To obtain ovarian follicles from sole females, the ovarian pieces were placed in Petri dishes containing 75% Leivovitz L-15 culture medium with L-glutamine and 100 μg/ml gentamicin at pH 7.5, and follicles at different developmental stages were manually isolated with watchmaker's forceps under an stereo microscope and frozen as above. Tissue biopsies from common sole were kindly provided by Dr. Marie-Laure Begout (IFREMER, France). The procedures relating to the care and use of fish and sacrifice were conducted in accordance with the protocols approved by the Ethics Committee (EC) of the Institut de Recerca i Tecnología Agroalimentàries (IRTA) following the European Union Council Guidelines (86/609/EU).

Frogs were purchased from the Centre de Ressources Biologiques Xénopes (University of Rennes, France) and maintained as previously described [10]. Oocytes were collected by surgical laparotomy from anesthetized females following a procedure approved by the Ethics

Committee for Animal and Human Experimentation (CEEAH) from Universitat Autònoma de Barcelona (Spain) and the Catalan Government (Direcció General de Polítiques Ambientals i Medi Natural; Project no. 10985).

## Reagents and antibodies

All reagents were purchased from Merck unless indicated otherwise. Antisera were raised in rabbits against synthetic peptides corresponding to the extracellular loop A of Senegalese sole Aqp1ab2 (GDRNNTYPDQEIKV), or to a specific amino acid sequence of the extracellular loop E of sole Aqp1ab2_v1 isoform (VSASRHSLTSRPSL) (Agrisera AB, Vännäs, Sweden). Antisera were affinity purified against the synthetic peptides. Previously characterized antibodies against gilthead seabream Aqp1ab1 [10], and commercial antibodies against α-tubulin (Merck #T9026), PDI (Merck #P7496), HA (Invitrogen #PA1-985) and FLAG (rabbit or mouse antibodies, Merck #F7425 and #F3165, respectively) were also employed.

## Identification and cloning of aquaporin splice forms

To identify *aqp1ab1* and *-1ab2* isoforms in seabream, halibut and sole we carried out RT-PCR employing species-specific oligonucleotide primers based on previously deposited sequences in GenBank (S1 Table) [10]. Total RNA was extracted from the ovary using the GenElute Total RNA Miniprep Kit and treated with Dnase I following the manufacturer instructions. The reverse transcription of RNA (5 μg) was performed with 0.5 μM oligo(dT)$_{12-18}$ (Thermo Fisher Scientific), 20 U of SuperScript II RT (Invitrogen), 5 × First-Strand Buffer (250 mM Tris-HCl, 375 mM KCl, 15 mM MgCl$_2$), 40 U of RNase OUT™ (Thermo Fisher Scientific), and 1 mM dNTPs (Promega), for 1.5 h at 42ºC. The PCR reaction was carried out with 1 μl of cDNA containing 1 × PCR buffer with Mg$^{2+}$, 0.2 mM dNTPs, 1 U Taq polymerase, and 0.2 μM of the forward and reverse oligonucleotide primers, in a 50-μl final volume. Reactions were amplified using one cycle at 95ºC for 2 min; 35 cycles at 95ºC for 30 sec, 60ºC for 30 sec, and 1 min at 72ºC; and 7 min for final elongation at 72ºC. The PCR products were run in 2% agarose gel, cloned into the pGEM-T Easy vector (Promega) and DNA Sanger sequenced (Macrogen). The full-length cDNAs of the splice variants identified were finally amplified by RT-PCR as described above, using primers located in the 5'- and 3'-end UTR regions of each paralog and the Easy-A high-fidelity PCR cloning enzyme (Agilent), and Sanger sequenced.

## Sequence and *in silico* structural analyses of aquaporin isoforms

Multiple nucleotide and amino acid sequence alignments were carried out using Clustal Omega (https://www.ebi.ac.uk/Tools/msa/clustalo/9) [38]. The 3-dimensional structures of the WT and variant channels were inferred using the model leverage option in the Modeler server (modbase.compbio. ucsf.edu), based on auto-selected AQP0 (2b6pA), AQP4 (3gd8A) and AQP5 (3d9sA) templates. The best scoring models were selected using the slow (Seq-Prf, position–specific iterative-basic local alignment search tool) assignment method, and rendered with MacPymol (https://pymol.org/2/).

## Microinjection of *X. laevis* oocytes and swelling assays

The cDNAs coding for the WT and splice forms were subcloned into the pT7Ts expression vector [39], in some cases fused with an HA or FLAG epitope tag at the C-terminus of the encoded proteins by using PCR. The cRNAs were synthesized *in vitro* with T7 RNA polymerase from *Xba*I-digested pT7Ts. The isolation and microinjection of oocytes, as well as the swelling assays, were carried out as previously described [40]. The oocytes were injected with

WT aquaporins (1 or 25 ng cRNA) or the different splice forms alone (25 or 50 ng cRNA), or with combinations of the WT with different amounts of the isoforms at ratios from 0.05 to 50 (for seabream and halibut Aqp1ab1) or from 0.004 to 1 (for sole Aqp1ab2). Oocytes expressing sole Aqp1ab2 and/or its isoforms were always co-injected with 25 ng of Atlantic halibut YwhazLb and treated with 100 μM FSK for 1 h prior to the swelling assays as previously described [10].

## Protein extraction

The total and plasma membrane fractions of *X. laevis* oocytes ($n$ = 10) were isolated as described previously [41]. Senegalese sole intact ovarian follicles or defolliculated oocytes ($n$ = 50) were homogenized in RIPA buffer (150 mM NaCl, 50 mM Tris pH 8.0, 1.0% Triton X-100, 0.5% sodium deoxycholate, 0.1% SDS, supplemented with 1 mM NaF, 1 mM $Na_3VO_4$, EDTA-free protease inhibitors, and 80 U benzonase). Samples were centrifuged at 14,000 × g for 10 min at 4ºC, and the supernatant mixed with 2 × Laemmli sample buffer and heated at 95˚C for 10 min. For defolliculation, follicles were incubated with 0.05% trypsin, 10 μM HCl, in L-15 medium during 2 h, followed by exposure to bovine serum albumin (BSA) diluted 1:500 in L-15 medium for 10 min, and a second treatment with 0.025% collagenase in L-15 for 30 min. Oocytes were subsequently washed with L-15, and stained with SYBR green to ensure total defolliculation.

## Co-immunoprecipitation

Frog oocytes expressing the different aquaporin constructs were homogenized in RIPA buffer as above, and an ~10% aliquot of the supernatant was collected as "input" and mixed with 2 × Laemmli sample buffer with protease inhibitors. The remaining supernatant was mixed with 50 μl of activated G-protein beads (PureProteome Protein A/G Mix Magnetic Beads, Millipore #LSKMAGAG), previously coupled to 5 μg of the corresponding primary antibody (either rabbit anti-seabream Aqp1ab1 or rabbit anti-FLAG), and incubated overnight with continuous mixing at 4ºC. After washing with PBS (137 mM NaCl, 2.7 mM KCl, 100 mM $Na_2HPO_4$, 2 mM $KH_2PO_4$, pH 7.4) with 0.05% Tween 20, the immunoprecipitate was eluted in 1 × Laemmli sample buffer, boiled at 95ºC for 10 min, and processed for immunoblotting.

## Western blotting

Laemmli-mixed protein extracts were subjected to 12% sodium dodecyl sulfate polyacrylamide gel electrophoresis (SDS-PAGE) and blotted onto Immuno-Blot nitrocellulose 0.2 μm membranes (Bio-Rad Laboratories, Inc.), as described previously [36]. The membranes were blocked for 1 h at room temperature in TBS (20 mM Tris, 140 mM NaCl, pH 7.6) with 0.1% Tween (TBST) containing 5% non-fat milk powder, and subsequently incubated with the primary antibodies (1:500 dilution) overnight at 4ºC. Bound antibodies were detected with horseradish peroxidase-coupled anti-rabbit IgG antibodies (Bio-Rad, #172–1019) for 1 h at room temperature. After washing in TBST, immunoreactive bands were revealed by the Immobilon™ Western chemiluminescent HRP substrate (Merck #WBKLS). In some extracts from sole ovarian follicles at different developmental stages, the intensity of the immunoreactive bands was scored by densitometry using the Quantity-One software (Bio-Rad Laboratories Inc.).

## Histology

Senegalese sole ovary biopsies were fixed in Bouin's solution (75 ml saturated aqueous solution of picric acid, 25 ml formalin, and 5 ml glacial acetic acid) overnight at room temperature.

Tissues were dehydrated and embedded in paraffin, and sections (8 μm) were attached to UltraStick/UltraFrost Adhesion slides (Electron Microscopy Sciences) and stained with hematoxylin and eosin, as previously described [36].

### Immunofluorescence microscopy

*X. laevis* oocytes and sole ovarian pieces were fixed for 6 h with 4% paraformaldehyde paraformaldehyde (PFA), dehydrated and embedded in paraplast. Sections of 7 μm in thickness were further attached to UltraStick/UltraFrost Adhesion slides and rehydrated before permeabilization with 0.1% Triton X-100 in PBS for 20 min. Sections were then blocked with 5% goat serum and 0.1% BSA in PBST (PBS with 0.1% Tween 20) for 1 h, and subsequently incubated with the primary antibodies diluted at 1:400 in PBS overnight at 4˚C in a humidified chamber. For the colocalization of the WT aquaporins and HA-tagged splice forms in *X. laevis* oocytes, SaAqp1ab1-WT was detected with a custom-made specific rabbit antibody [10], while FLAG-tagged HhAqp1ab1-WT and SseAqp1ab2-WT were detected with a rabbit or mouse anti-FLAG antibody. All the splice forms were detected with a goat anti-HA antibody. For the colocalization of the channels with PDI, we combined the previous antibodies with a rabbit anti-PDI. After washing with PBS, the sections were incubated with a sheep anti-rabbit IgG coupled with Cy3 (Merck #C2306), and with either goat anti-mouse IgG Alexa Fluor 488 (Invitrogen #A-11001) or donkey anti-goat IgG Alexa Fluor 488 (Invitrogen # A-11055), both diluted at 1:800 in PBS for 1 h. To colocalize PDI with SaAqpa1ab1-WT, we first performed PDI detection with the rabbit anti-PDI and the sheep anti-rabbit Cy3-coupled secondary antibody, and after fixing with PFA for 1 h, slides were incubated with the SaAqp1ab1 antibody previously labeled with Zenon™ Rabbit IgG Alexa Fluor™ 488 Labeling Kit (Invitrogen, #Z25302) following manufacturer instructions. After 1 h, sections were washed and fixed again with PFA for 15 min. In some experiments, the plasma membrane and nuclei were counterstained with WGA Alexa Fluor® 647 conjugate (Life Technologies Corp. #W32466, dilution 1: 10000) and DAPI (Merck #G8294, dilution 1: 5000), respectively. Control sections for the immunolocalization of Aqp1ab2 and Aqp1ab2_v1 in sole ovaries were incubated with antibodies previously preadsorbed with 5-fold excess of their respective immunizing peptides for 1 h at 37ºC. Finally, all sections were mounted with fluoromount aqueous mounting medium (Merck #F4680), and immunofluorescence was observed and documented with a Zeiss Axio Imager Z1/Apo-Tome fluorescence microscope (CarlZeiss Corp.).

### Statistics

All statistical analyses were performed with GraphPad Prism v10.0.2 (232) (GraphPad Software). Comparisons between two independent groups were made by the two-tailed unpaired Student's *t*-test, or by the nonparametric Mann Whitney U test in case of non normal distribution. The statistical significance among multiple groups was analyzed by one-way ANOVA, followed by the Tukey's multiple comparison test, or by the non-parametric Kruskal-Wallis test and further Dunn's test for nonparametric *post hoc* comparisons. Statistical significance was accepted at $p \leq 0.05$. Data are expressed as mean ± standard error of means (SEM).

### Supporting information

**S1 Fig. Tagging HhAqp1ab1 and SsAqp1ab2 constructs do not affect channel function.** $P_f$ of *X. laevis* oocytes injected with water (W, controls), or expressing non-tagged or Flag-tagged Atlantic halibut HhAqp1ab1-WT or sole SsAqp1ab2-WT. Oocytes injected with SsAqp1ab2-WT were co-injected with non-tagged halibut YwhazLb and exposed to FSK. The data

are the mean ± SEM ($n$ = 12 oocytes per treatment, indicated with dots above each bar).
(TIF)

**S2 Fig. Immunostaining controls of Senegalese sole ovarian follicles at different developmental stage.** Histological sections were incubated with Aqp1ab2-Nt and Aqp1ab2_v1 antisera preabsorbed with the immunizing peptides, indicating the specificity of the antibodies. Abbreviations: o, oocyte; y, yolk globule; gv, germinal vesicle; ve, vitelline envelope; cp, capillary; ca, cortical alveoli. Scale bars, 10 μm.
(TIF)

**S1 File. Raw RT-PCR gels images.**
(TIF)

**S2 File. Raw Western blot images from Fig 5D–5F.**
(TIF)

**S3 File. Raw Western blot images from Fig 5G–5I.**
(TIF)

**S4 File. Raw Western blot images from Fig 6B, 6D and 6F.**
(TIF)

**S5 File. Raw Western blot images from Fig 8.**
(TIF)

**S6 File. Raw data.**
(XLS)

**S1 Table. Oligonucleotide primers employed for the RT-PCR screening of splice forms expression of *aqp1ab*-type genes in teleosts.**
(DOC)

## Author Contributions

**Conceptualization:** Joan Cerdà.

**Formal analysis:** Alba Ferré, François Chauvigné, Cinta Zapater, Roderick Nigel Finn, Joan Cerdà.

**Funding acquisition:** Joan Cerdà.

**Investigation:** Alba Ferré, François Chauvigné, Cinta Zapater.

**Supervision:** Joan Cerdà.

**Writing – original draft:** Alba Ferré.

**Writing – review & editing:** Roderick Nigel Finn, Joan Cerdà.

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
