## [Decision Letter · Decision Letter 0]

17 Oct 2023

PONE-D-23-28817Aquaporin splice variation differentially modulates channel function during marine teleost egg hydrationPLOS ONE

Dear Dr. Cerdà,

Thank you for submitting your manuscript to PLOS ONE. After careful consideration, we feel that it has merit but does not fully meet PLOS ONE’s publication criteria as it currently stands. Therefore, we invite you to submit a revised version of the manuscript that addresses the points raised during the review process.

We look forward to receiving your revised manuscript.

Kind regards,

Michael Schubert

Academic Editor

PLOS ONE

Journal Requirements:

 "This work was supported by the Spanish Ministry of Science and Innovation (MCIN/AEI/ 10.13039/501100011033), the European Regional Development Fund (ERDF) “A way of making Europe” (European Union), Grant no. AGL2016-76802-R (to J.C.), and the Norwegian Research Council (RCN) Grant no. 294768/E40 (to R.N.F). A.F. was recipient of a predoctoral contract from Spanish MCIN (BES-2014-068745). R.N.F. was supported by the University of Bergen (Norway)."

Reviewers' comments:

Reviewer's Responses to Questions

**Comments to the Author**

1. Is the manuscript technically sound, and do the data support the conclusions?

Reviewer #1: Yes

Reviewer #2: Yes

2. Has the statistical analysis been performed appropriately and rigorously? 

Reviewer #1: No

Reviewer #2: Yes

3. Have the authors made all data underlying the findings in their manuscript fully available?

Reviewer #1: Yes

Reviewer #2: Yes

4. Is the manuscript presented in an intelligible fashion and written in standard English?

Reviewer #1: Yes

Reviewer #2: Yes

5. Review Comments to the Author

Reviewer #1: This informative and well presented MS capably documents the subtypes and patterns of expression of aquaporin channels during developmental changes in oocyte hydration during meiosis in 3 species of marine teleosts. The marine species used (Ss sole, Sa seabream, Hh halibut) have eggs that float in seawater and show conserved genes for Aqp1ab1 and -1ab2 channels, in contrast to species that carry eggs to maturation internally. This pattern is consistent with a role for these channels in the oocyte hydration process.

Figs 6 and 7 show a thorough analysis of patterns of SsAqp1 expression during early development in oocytes and follicular cells, and evidence for polarization by cortical accumulation. Appropriate controls were included to confirm specificity of the custom generated antibodies. The images are of excellent quality. In Fig 8 the Western blot data are clear, convincing and well presented.

Overall this MS is a comprehensive and capably presented body of work that provides detailed insight into patterns of expression of Aqp channel subtypes. Future work determining the functional roles of the teleost Aqp channels which don't appear to function as water channels will be of interest. The authors might consider the precedent set by another developmentally important member of the Aqp family known as Big Brain, essential for Drosophila neurogenesis.

Points that merit further consideration are listed below:

1. In Fig. 3, the term "Functional characterization" only refers to the first three panels. The remaining panels illustrate localization but do not address function per se. A more complete set of data would be helpful, showing average oocyte volume as a function of time in hypotonic saline as the raw data used to calculate Pf values. Data from swelling assays with the Flag- and HA-tagged constructs are needed to show that the C-terminal modification did not impair full channel function, which has been observed in other Aqp classes. As well, SsAQP1 responses with and without forskolin would be helpful for the stated purpose of providing a characterization of the responses.

Clarification of the n value is needed, which states 24 oocytes were used in two experiments. Is this 24 for every treatment group and construct, or 24 total for all assays combined? Ideally a minimum of 10 oocytes per treatment group would be expected. n values could be added near each histogram bar as was done in Fig 4. Statistical analyses with T tests alone are insufficient. An initial analysis of variance for all groups, and and an assessment of normal distributions would be needed before post-hoc T test comparisons. Immunolabeled images clearly show the predominant ER localization, and membrane expression of the Sa wt and selected variants in Hh and Ss.

2. In Fig 4, the SsAQP1 is shown to be incredibly sensitive to low levels of splice variant coexpression, but this construct was uniquely tested in the presence of YwhazLg and FSK. Yhwaz genes participate as master regulators of intracellular signaling. Differential phosphorylation of Aqp1ab2 regulates trafficking to proximal versus distal microvilli. The authors noted that Aqp1ab2 will traffic to plasma membrane only when it is bound to YwhazLb protein in the presence of forskolin. Yet despite this key observation, the roles of Yhwaz-like proteins in Aqp responses are not systematically addressed in this study. It would be valuable to determine it the sensitivity of the other constructs can also be enhanced in the presence of these co-factors. This additional information could be limited to a selected ratio which produced near half-maximal reduction of Pf in order to enable a clear comparison of the functional properties of the channels and variants. The dose of FSK is given as 100 µM in Methods, but it would be good to also specify the duration of exposure in advance of the swelling assay.

3. Dominant-negative effects were greatest for SsAqp1ab2, followed by HhAqp1ab1_v1, and then the 40 fold less effective SaAqp1ab1_v1 and SaAqp1ab1_v2 variants. The differential ability of variants to interfere might be limited by HA epitope tagging effects on the efficiency of protein trafficking to the plasma membrane. Immunostaining in Fig. 5 showed relatively poor membrane localization of SaAQP1 and HhAqp1, as compared to SsAqp1 for which PM localization appeared convincing.

Minor details:

--Fig 1G. line 867. The two splice forms of aqp1ab2 were seen not only in ovary but also in testis in sole.

--Fig 2. line 870. "Structure... of splice variants" in legend title could be restated more accurately as the 'theoretical structures' or 'model-based structures' since the diagrams are based on modeling, not directly on crystallography data.

Line 874. The green-blue color is cyan, rather than Cyagen which is a biotech trade name.

Lines 875, 992. The abbreviation NPA has not been defined in text or legends.

Reviewer #2: General comments

The authors have found new mRNA isoforms of teleost Aqp1ab channels by PCR technologies and characterized these variants using heterologous expression systems by Xenopus laevis and Senegalese sole. In-frame intronic splice variants of Aqp1ab1 caused dominant-negative inhibition of the canonical channel function through hetero-oligomerization and retention in the ER. On the other hand, Aqp1ab2-type variants dominantly negatively inhibited water flux. Studies by ovarian follicular expression showed that the Aqp1ab2 isoforms had a function for water retention of the eggs when the

canonical channel was recycled. These findings may represent a new regulatory mechanism by isoforms of Aqp1ab1 and Aqp1ab2 in marine teleost aquaporin membrane expressions, and this paper is interesting. This referee has minor comments.

Minor comments

1. Figures 3 and 5: The figures in the downloaded paper were not clear and should be improved.

P14, L446: There is no period.

P14, L468: This referee would like the authors to explain more about “Avp signaling”.

6. PLOS authors have the option to publish the peer review history of their article (what does this mean?). If published, this will include your full peer review and any attached files.

Reviewer #1: **Yes: **Andrea Yool

Reviewer #2: No

---

## [Author Response · Author response to Decision Letter 0]

8 Nov 2023

Journal Requirements:

The manuscript has been formatted as indicated.

This information has been added in Materials and Methods section.

This information is provided in the first page of the manuscript.

 "This work was supported by the Spanish Ministry of Science and Innovation (MCIN/AEI/ 10.13039/501100011033), the European Regional Development Fund (ERDF) “A way of making Europe” (European Union), Grant no. AGL2016-76802-R (to J.C.), and the Norwegian Research Council (RCN) Grant no. 294768/E40 (to R.N.F). A.F. was recipient of a predoctoral contract from Spanish MCIN (BES-2014-068745). R.N.F. was supported by the University of Bergen (Norway)."

The statement has been added in the manuscript.

Uncropped gels and immunoblot images are available in Supporting Information (files S1-S5).

The reference list is complete and correct.

Reviewer #1: 

1. In Fig. 3, the term "Functional characterization" only refers to the first three panels. The remaining panels illustrate localization but do not address function per se. A more complete set of data would be helpful, showing average oocyte volume as a function of time in hypotonic saline as the raw data used to calculate Pf values. Data from swelling assays with the Flag- and HA-tagged constructs are needed to show that the C-terminal modification did not impair full channel function, which has been observed in other Aqp classes. As well, SsAQP1 responses with and without forskolin would be helpful for the stated purpose of providing a characterization of the responses.

According to the comments of the reviewer, the title of the legend for Fig. 3 has been changed as “Functional characterization and subcellular localization of teleost wild-type Aqp1ab1 and -1ab2 and their splice variants”.

We think however that the oocyte volume data would be somehow redundant with the values of Pf (and will take more space in the figure) and are not necessary. Also note that the individual values of Pf for all the experiments in the manuscript are shown in Supplementary Information.

In the initial manuscript, all the functional experiments in oocytes (Fig. 3 and 4) were carried out using untagged constructs. This was not well indicated in the manuscript and is now specifically stated in the Results section and in the figure legends. For the subcellullar localization, Co-IP and immunoblotting experiments, we used tagged constructs. The immunostaining images in Figure D-F suggested that the tag did not affect the trafficking of the canonical channels to the oocyte plasma membrane. To demonstrate this, we have carried out additional experiments which confirm that the Flag tag does not affect the Pf of oocytes (S1 Fig).

A comprehensive characterization of the response of teleost Aqp1aa, Aqp1ab1 and Aqp1ab2 to FSK, including the identification of the amino acid residues involved in each channel, has been provided in ref. [10], which is extensively cited in the manuscript.

Clarification of the n value is needed, which states 24 oocytes were used in two experiments. Is this 24 for every treatment group and construct, or 24 total for all assays combined? Ideally a minimum of 10 oocytes per treatment group would be expected. n values could be added near each histogram bar as was done in Fig 4. Statistical analyses with T tests alone are insufficient. An initial analysis of variance for all groups, and and an assessment of normal distributions would be needed before post-hoc T test comparisons. Immunolabeled images clearly show the predominant ER localization, and membrane expression of the Sa wt and selected variants in Hh and Ss.

In that specific experiment, the number of oocytes was 24 per construct and therefore our statistical analysis is correct. The number of oocytes is now indicated above each bar in panels 3A-C to avoid confusion. Please, note that as stated in Materials and Methods of the previous version of the manuscript, the assessment of normal distributions among the groups was carried out and nonparametric post hoc comparisons were applied when needed. We have stated this in the revised version (lines 670-677).

2. In Fig 4, the SsAQP1 is shown to be incredibly sensitive to low levels of splice variant coexpression, but this construct was uniquely tested in the presence of YwhazLg and FSK. Yhwaz genes participate as master regulators of intracellular signaling. Differential phosphorylation of Aqp1ab2 regulates trafficking to proximal versus distal microvilli. The authors noted that Aqp1ab2 will traffic to plasma membrane only when it is bound to YwhazLb protein in the presence of forskolin. Yet despite this key observation, the roles of Yhwaz-like proteins in Aqp responses are not systematically addressed in this study. It would be valuable to determine it the sensitivity of the other constructs can also be enhanced in the presence of these co-factors. This additional information could be limited to a selected ratio which produced near half-maximal reduction of Pf in order to enable a clear comparison of the functional properties of the channels and variants. The dose of FSK is given as 100 µM in Methods, but it would be good to also specify the duration of exposure in advance of the swelling assay.

A systematic study of the binding of YwhazL proteins to Aqp1ab1 and Aqp1ab2 channels in teleosts has been published in ref. [10]. This study shows that Aqp1ab1 binds preferentially to YwhazLa to enhance trafficking, whereas Aqp1ab2 interacts exclusively with YwhazLb. We therefore agree with the reviewer that the dose response of the inhibition of the Aqp1ab splice forms on the canonical channels should also be tested in the presence of YwhazLa. To address this, we have performed new experiments using Aqp1ab1 splice forms at concentrations that produce half-maximal reduction of Pf in oocytes expressing SaAqp1ab1 or HhAqp1ab1, in the presence or absence of YwhazLa. In these experiments, oocytes were exposed to FSK, since the trafficking of these channels with the coexpression of YwhazLa is only enhanced after PKA activation (see ref. [10]). The data confirm that the sensitivity of SaAqp1ab1 or HhAqp1ab1 to the splice variants increased, suggesting that Aqp1ab1 channels are possibly highly sensitive to low levels of splice variant coexpression under in vivo conditions. These new data are now shown in Fig. 4D and explained in the Results section.

The duration of FSK treatment is now specifically indicated in Results and figure legends.

3. Dominant-negative effects were greatest for SsAqp1ab2, followed by HhAqp1ab1_v1, and then the 40 fold less effective SaAqp1ab1_v1 and SaAqp1ab1_v2 variants. The differential ability of variants to interfere might be limited by HA epitope tagging effects on the efficiency of protein trafficking to the plasma membrane. Immunostaining in Fig. 5 showed relatively poor membrane localization of SaAQP1 and HhAqp1, as compared to SsAqp1 for which PM localization appeared convincing.

As mentioned above, all Pf measurements were carried out using non-tagged constructs. Also, note that the HA tag does not affect the binding of the YwhazL proteins to the Aqp1ab channels as we previously showed in ref. [10].

Minor details:

--Fig 1G. line 867. The two splice forms of aqp1ab2 were seen not only in ovary but also in testis in sole.

The expression in testis has been mentioned in the figure legend.

--Fig 2. line 870. "Structure... of splice variants" in legend title could be restated more accurately as the 'theoretical structures' or 'model-based structures' since the diagrams are based on modeling, not directly on crystallography data.

Corrected as suggested by the reviewer.

Line 874. The green-blue color is cyan, rather than Cyagen which is a biotech trade name.

Corrected.

Lines 875, 992. The abbreviation NPA has not been defined in text or legends.

The NPA abbreviation is now defined in the legend of Fig. 2.

Reviewer #2: 

Minor comments

1. Figures 3 and 5: The figures in the downloaded paper were not clear and should be improved.

The final figures are in TIFF with high resolution. The pdf copy did not show the images with the correct resolution. In any case, we have submitted revised TIFF files for Figs. 2, 3, 4, 5, 6 and 7 with improved resolution.

P14, L446: There is no period.

Corrected.

P14, L468: This referee would like the authors to explain more about “Avp signaling”.

The control of Aqp1ab1 and Aqp1ab2 trafficking by Avp-Avpr2aa-PKA signaling is extensively explained in our recently published paper (ref. [13]). We have not included a further discussion of this mechanism to keep the manuscript streamlined in order to better communicate the significance of the present results.

---

## [Editor Report · Decision Letter 1]

9 Nov 2023

Aquaporin splice variation differentially modulates channel function during marine teleost egg hydration

PONE-D-23-28817R1

Dear Dr. Cerdà,

We’re pleased to inform you that your manuscript has been judged scientifically suitable for publication and will be formally accepted for publication once it meets all outstanding technical requirements.

Kind regards,

Michael Schubert

Academic Editor

PLOS ONE

---

## [Editor Report · Acceptance letter]

14 Nov 2023

PONE-D-23-28817R1 

Aquaporin splice variation differentially modulates channel function during marine teleost egg hydration 

Dear Dr. Cerdà:

I'm pleased to inform you that your manuscript has been deemed suitable for publication in PLOS ONE. Congratulations! Your manuscript is now with our production department. 

Kind regards, 

on behalf of

Dr. Michael Schubert 

Academic Editor

PLOS ONE